# DATA IMPUTATION BY PURSUING BETTER CLASSIFICATION: A SUPERVISED LEARNING APPROACH

## ABSTRACT

Data imputation, the process of filling in missing feature elements for incomplete data sets, plays a crucial role in data-driven learning. A fundamental belief is that data imputation is helpful for learning performance, and it follows that the pursuit of better classification can guide the data imputation process. While some works consider using label information to assist in this task, their simplistic utilization of labels lacks flexibility and may rely on strict assumptions. In this paper, we propose a new framework that effectively leverages supervision information to complete missing data in a manner conducive to classification. Specifically, this framework operates in two stages. Firstly, it leverages labels to supervise the optimization of similarity relationships among data, represented by the kernel matrix, with the goal of enhancing classification accuracy. To mitigate overfitting that may occur during this process, a perturbation variable is introduced to improve the robustness of the framework. Secondly, the learned kernel matrix serves as additional supervision information to guide data imputation through regression, utilizing the block coordinate descent method. The superiority of the proposed method is evaluated on four real-world data sets by comparing it with state-of-the-art imputation methods. Remarkably, our algorithm significantly outperforms other methods when the data is missing more than 60% of the features.

## 1 INTRODUCTION

The presence of missing data poses significant challenges in machine learning and data analysis (García-Laencina et al., 2010; Little & Rubin, 2019; Emmanuel et al., 2021). In a given sequence of triplets $\{(\boldsymbol{x}_i, \boldsymbol{o}_i, y_i)\}_{i=1}^N$, where $\boldsymbol{x}_i \in \mathbb{R}^d$ and $y_i \in \mathbb{R}$, we typically use the set of observable features $o_i \subseteq 2^{\{1,\cdots,d\}}$ and missing features $*$ to represent the missing data $x_{o_i}^i \in (\mathbb{R} \cup \{*\})^d$. This sequence is then consolidated into an incomplete data set $\mathcal{D} = \{x_{o_i}^i, y_i\}_{i=1}^N$. The presence of these missing entries can be attributed to various reasons. Firstly, it may occur due to limitations in data collection, such as time constraints or limited resources, resulting in certain observations being omitted. Secondly, data can be intentionally designed to have missing values, for instance, in survey studies where participants may choose not to answer certain questions, thereby introducing missing values into the data set. By completing missing data, researchers can uncover the underlying structure and relationships within the data set, optimizing the utilization of the complete data set to enhance the performance and efficiency of subsequent analyses. Therefore, the significance of filling in missing data becomes self-evident.

The existing data imputation methods primarily focus on the relationships among features. For example, mean imputation (MI, (Schafer, 1997)) considers the mean value of the feature, while other methods consider the low-rank property of the imputed matrix (Sheikholesalmi et al., 2014; Xu et al., 2020) or incorporate similarity information (Troyanskaya et al., 2001; Batista & Monard, 2002; Śmieja et al., 2019). Some methods also utilize label information (Smola et al., 2005; Allison, 2009; Goldberg et al., 2010). However, the effective utilization of label information has not been fully explored yet. Considering a fundamental observation that *data are helpful for distinguishing the labels*, we can derive an imputation criterion:

*data imputation should lead to improved classification performance.*

Thus we expect imputation results that can perform better on subsequent tasks. Let us consider a toy example illustrated in Figure 1. In the synthetic two moons data set, we have a positive data with

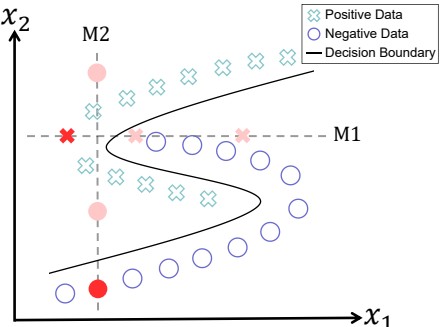

Figure 1: Imputation of the two moons data set. Positive and negative data are denoted by X and O, while the solid curve denotes the ideal decision boundary of the classifier. The gray dashed lines M1 and M2 represent all possible imputations for a positive data with a missing $x_1$ dimension and a negative data with a missing $x_2$ dimension, respectively. Among them, the red results are more likely to lead to better outcomes in the subsequent classification task compared to the pink results, making them more desirable.

a missing value in the $x_1$ dimension, and a negative data with a missing value in the $x_2$ dimension. All possible imputations for these two data are represented by gray dashed lines, labeled as M1 and M2. By applying the proposed imputation criterion, we prioritize selecting the red results over the pink results to improve the subsequent classification task.

The task now becomes to find suitable missing values that aid in classification. Classification relies on the relationships among the data, which are captured by kernel functions in kernel-based learning algorithms, e.g., support vector machine (SVM, (Vapnik, 1999; Schölkopf & Smola, 2002; Steinwart & Christmann, 2008)). Specifically, a kernel matrix or Gram matrix, denoted as $\boldsymbol{K} \in \mathbb{R}^{N \times N}$, is constructed, where $K_{i,j} = k(\boldsymbol{x}_i, \boldsymbol{x}_j)$ represents the similarity between data $\boldsymbol{x}_i$ and $\boldsymbol{x}_j$ using a proper positive definite kernel function $k(\cdot, \cdot) : \mathbb{R}^d \times \mathbb{R}^d \rightarrow \mathbb{R}$. A classifier is then trained based on this kernel matrix $\boldsymbol{K}$. When handling missing data, represented as $\boldsymbol{x}_{\boldsymbol{o}_i}^i$, the value of $K_{i,\cdot}$ is unknown. Thus, it is necessary to find $\tilde{K}_{i,j}$ for data with missing features in order to establish appropriate relationships with other data. We will accomplish this by improving the prediction accuracy of the subsequent classification task while considering prior knowledge of the missing values and imposing constraints on the adjustment of the kernel matrix. The next step is to recover the missing values from the imputed matrix $\tilde{\boldsymbol{K}}$. This step is relatively easy, particularly in the traditional scenario where $N \gg d$. Because for each data, there is $N-1$ available supervised information stored in the kernel matrix to guide the imputation of $d$ or fewer elements. In this paper, we propose a two-stage learning framework for data imputation: I) the kernel matrix is completed by pursuing high classification accuracy; II) the missing features are reconstructed to approximate the optimized kernel matrix. With the proposed method, we can find good estimates for missing values that lead to better classification performance in the subsequent task. Therefore, when faced with situations like the one shown in Figure 1, we are able to obtain the imputation results indicated by the red crosses and circles.

In the first stage, specifically, we begin by computing the initial kernel matrix based on the observed features. This matrix is then further modified through a process referred to as *kernel matrix imputation* throughout this paper. Recent studies (Liu et al., 2020; Wang et al., 2023) have demonstrated that integrating kernel learning and classifier optimization tasks, and alternating between them, can improve classification performance and naturally lead to a fine-tuned kernel matrix. Inspired by these ideas, we adopt a similar approach by combining the tasks of kernel matrix imputation and classifier training into a maximum-minimum optimization problem. However, we encounter a challenge during the kernel matrix imputation process. The flexibility of modifying the similarity relationships between data within our framework can make the classifier susceptible to overfitting. To address this issue, we introduce an additional perturbation matrix during the optimization of the kernel matrix, thereby enhancing the classifier's robustness. In the second stage, we utilize the block coordinate descent (BCD, (Tseng, 2001)) method to solve a non-convex problem. Although find-

ing the global optimal solution is challenging, our numerical experiments demonstrate that when the data size is much larger than the feature dimension, the abundant supervisory information from the kernel matrix effectively guides the imputation process. As a result, we can still achieve highly accurate solutions. The main contributions of this paper are summarized as follows:

- To fully leverage label information, we propose a novel two-stage data imputation framework. This framework optimizes the similarity relationships between data by utilizing supervision information and guides the data imputation process accordingly.

- We have developed a nonparametric method to impute the kernel matrix, which is performed alternately with the training of the classifier. By introducing a perturbation variable, we enhance the robustness of the classifier.

- We provide a solving algorithm based on the BCD to accurately recover missing data from a given kernel matrix. This algorithm effectively utilizes the learned similarity information among the training data.

- Experimental results on real data sets indicate that the imputed data generated by our framework exhibits excellent performance in classification tasks. Notably, when the missing ratio exceeds 60%, our algorithm outperforms other imputation methods significantly.

## 2 RELATED WORK

The task of learning a classifier from data with missing values has been extensively studied in the past few decades, leading to the development of three main approaches as below.

**Filling Data Before Classification.** A large portion of the research focuses on filling the data before using it in the subsequent classification task. For instance, Ghahramani & Jordan (1993) introduces a framework based on maximum likelihood density estimation for coping with missing data, utilizing mixture models and the expectation-maximization principle. And Bhattacharyya et al. (2004) modeled the missing values as Gaussian random variables and estimated their mean and covariance from the observed data and labels, which were then used in a modified SVM. Later, Williams et al. (2007) used a Gaussian mixture model to estimate the conditional probability of missing values given the observed data and imputed them using expectations. Additionally, some work estimated complete data by computing the corresponding marginal distributions of the missing data (Smola et al., 2005) or assumed a low-rank subspace for the data and treated labels as an additional feature to handle missing values (Goldberg et al., 2010). But these methods separate the imputation process from the classification task, which may not necessarily lead to improved classification performance with their imputation outcomes.

**Filling the kernel matrix Before Classification.** Another research direction focuses on kernel-based machine learning algorithms. In this approach, classification or regression tasks only require the kernel matrix based on the training data. Therefore, methods have been explored to compute kernel function values between data with missing values instead of directly imputing the missing data. Hazan et al. (2015) proposed a parameterized kernel function based on the low-rank assumption of data, which allows for input vectors with missing values. However, when computing similarities, only the dimensions present in both data are considered, and the rest are discarded. There is also work that modeled the squared distance between two missing data as a random variable following a gamma distribution and computed the expectation of the Gaussian kernel function under missing data (Mesquita et al., 2019). Similarly, Śmieja et al. (2019) modeled possible outcomes of missing values with the data distribution and computed the expectation of similarity between two missing data. Still, these methods have limitations in flexibly utilizing the supervision information from labels during the imputation process, which consequently restricts their performance in practical tasks.

**Direct Classification of Incomplete Data.** Furthermore, researchers have explored methods for directly classifying missing data. In the study by Pelckmans et al. (2005), they defined a modified risk to address the uncertainty in prediction results caused by missing data. And in (Chechik et al., 2008), they maximized the margin of the subspace correlated with the observed data. However, when computing kernel values, they encounter similar issues of incomplete information utilization as in (Hazan et al., 2015). In a different approach, Dekel et al. (2010) treated missing values as a specific type of noise and developed a linear programming problem that resembled SVM formulation in

order to learn a robust binary classifier. And some work optimize the nonlinear classifier while seeking the linear subspace where the data might lie (Sheikholesalmi et al., 2014; Xu et al., 2020). Due to the necessity of predefining the dimension of the subspace in these methods, they may fail to fully capture the intrinsic structure of the data, thus constraining its flexibility in practical applications. Recently, a neural network-based method called NeuMiss was introduced (Le Morvan et al., 2020; 2021). NeuMiss utilized a Neumann-series approximation of the optimal predictor and demonstrated robustness against various missing data mechanisms. In addition to the aforementioned algorithmic work that are more application-oriented, there are also efforts dedicated to theoretical derivations related to missing data. For example, Bullins et al. (2016) provided the information-theoretic upper and lower bounds of precision limits for vanilla SVM in the context of learning with missing data. Josse et al. (2019) proved that a predictor designed for complete observations can achieve optimal predictions on incomplete data by utilizing multiple imputation.

# 3 TWO-STAGE DATA IMPUTATION

## 3.1 PRELIMINARIES

**Notations**. The set of real numbers is written as $\mathbb{R}$. The set of integers from 1 to $N$ is written as $[N]$. The cardinality of the set $\mathcal{A}$ is written as $|\mathcal{A}|$ denotes. We take $a$, $\boldsymbol{a}$, and $\boldsymbol{A}$ to be a scalar, a vector, and a matrix, respectively. Let $\mathbf{0}$ and $\mathbf{1}$ denote vectors consisting of all zeros and all ones with the appropriate size. The inner product of two vectors in the given space is written as $\langle \cdot, \cdot \rangle$. We take $\mathrm{diag}(\boldsymbol{a})$ to be an operator that extends a vector to a diagonal matrix. The Frobenius norm of a matrix is written as $\| \cdot \|_\mathrm{F}$. The set of positive semi-definite (PSD) matrices is written as $\mathcal{S}_+$. The Hadamard product is written as $\odot$.

We begin by introducing the vanilla SVM and then demonstrate how to fill in missing data to achieve improved classification performance. In the hard-margin SVM, the objective is to find a hyperplane $\boldsymbol{w}^\top \boldsymbol{x} + b = 0$ that maximally separates the data $\boldsymbol{x} \in \mathbb{R}^d$ of different classes with zero training errors. When the data is not linearly separable, the soft-margin SVM is proposed, which allows for some training errors by introducing slack variables denoted as $\xi_i \geq 0$ for each training sample. Meanwhile, in order to find a more flexible decision boundary, a nonlinear mapping $\phi(\cdot)$ from the $\mathbb{R}^d$ to a reproducing kernel Hilbert space $\mathcal{H}_k$ is introduced, yielding the decision function $f(\boldsymbol{x}) = \mathrm{sign}(\boldsymbol{w}^\top \phi(\boldsymbol{x}) + b)$. With the completed data, the optimization problem can be formulated as

$$
\begin{aligned}
\min_{\boldsymbol{w}, b, \{\xi_i\}} \quad & \frac{1}{2} \|\boldsymbol{w}\|_2^2 + C \sum_{i=1}^N \xi_i \\
\text{s.t.} \quad & y_i(\boldsymbol{w}^\top \phi(\boldsymbol{x}_i) + b) \geq 1 - \xi_i, \\
& \xi_i \geq 0, \ \forall i \in [N],
\end{aligned}
\tag{1}
$$

where $C \geq 0$ is a hyperparameter that controls the balance between maximizing the margin and minimizing the training errors. For problem (1), previous researchers have proven that we only need to solve its corresponding dual problem:

$$
\begin{aligned}
\max_{\boldsymbol{\alpha} \in \mathbb{R}^N} \quad & \mathbf{1}^\top \boldsymbol{\alpha} - \frac{1}{2} \boldsymbol{\alpha}^\top \mathrm{diag}(\boldsymbol{y}) \boldsymbol{K} \mathrm{diag}(\boldsymbol{y}) \boldsymbol{\alpha} \\
\text{s.t.} \quad & \boldsymbol{y}^\top \boldsymbol{\alpha} = 0, \ \mathbf{0} \leq \boldsymbol{\alpha} \leq C\mathbf{1},
\end{aligned}
\tag{2}
$$

where $\boldsymbol{y} = [y_1; y_2; \ldots; y_N] \in \mathbb{R}^N$ and each entry $K_{i,j} = \langle \phi(\boldsymbol{x}_i), \phi(\boldsymbol{x}_j) \rangle_{\mathcal{H}_k}$ of the kernel matrix $\boldsymbol{K} \in \mathbb{R}^{N \times N}$ represents the similarity between two data. Through a technique known as the kernel trick, this similarity can be computed using a predefined positive definite kernel function $k(\boldsymbol{x}_i, \boldsymbol{x}_j) : \mathbb{R}^d \times \mathbb{R}^d \to \mathbb{R}$, allowing us to calculate it without knowing the explicit expressions of $\phi$. And the resulting kernel matrix $\boldsymbol{K}$ is guaranteed to be PSD.

## 3.2 STAGE I: KERNEL MATRIX IMPUTATION WITH SVM

In this stage, our goal is to find a new kernel matrix $\tilde{\boldsymbol{K}}$ that further reduces the objective function value in (2), aiming to improve the classification accuracy of the trained classifier. We propose a novel algorithm that integrates kernel matrix imputation with the classification task. Initially, we

compute the observed kernel matrix $K_o$, and then optimize an adjustment matrix $K_\Delta$ with the same dimension to derive $\tilde{K} = K_o \odot K_\Delta$ as the optimized kernel matrix. This approach distinguishes itself from previous algorithms by effectively utilizing each observed feature value. Considering the Gaussian kernel function $k_\gamma(\boldsymbol{x}, \boldsymbol{y}) = \exp(-\gamma\|\boldsymbol{x} - \boldsymbol{y}\|_2^2)$ as an example and recalling the definition of the incomplete data set $\mathcal{D} = \{\boldsymbol{x}_{\boldsymbol{o}_i}^i, y_i\}_{i=1}^N$, we can decompose $K_{i,j}$ into an observed part and unknown parts:

$$
\begin{aligned}
K_{i,j} &= \exp\left(-\gamma D_{i,j}\right) \text{ where} \\
D_{i,j} &= \|\boldsymbol{x}_{\boldsymbol{o}_i}^i - \boldsymbol{x}_{\boldsymbol{o}_j}^j\|_2^2 \\
&= \sum_{p\in\boldsymbol{o}_i\cap\boldsymbol{o}_j} \left(x_p^i - x_p^j\right)^2 + \sum_{p\in\boldsymbol{o}_i\backslash\boldsymbol{o}_j} \left(x_p^i - *\right)^2 + \sum_{p\in\boldsymbol{o}_j\backslash\boldsymbol{o}_i} \left(* - x_p^j\right)^2 + \sum_{p\notin\boldsymbol{o}_i\cup\boldsymbol{o}_j} \left(* - *\right)^2.
\end{aligned}
\tag{3}
$$

Recall that $*$ represents an unknown real number. Methods in (Chechik et al., 2008; Hazan et al., 2015) only consider the similarity between the intersecting observed features of two data, i.e., the part related to $p \in \boldsymbol{o}_i \cap \boldsymbol{o}_j$, disregarding features such as $x_p^i : p \in \boldsymbol{o}_i\backslash\boldsymbol{o}_j$. In contrast, our algorithm calculates $\boldsymbol{K}_o$ using the first term in the last line of (3), and implicitly utilize features from the second and third terms by constraining the potential ranges for $\boldsymbol{K}_\Delta$. Assuming $x$ is normalized within the range of $[0, 1]$, the ranges of the second, third, and fourth terms in (3) are denoted as $[0, \sum_{p\in\boldsymbol{o}_i\backslash\boldsymbol{o}_j} \max\{(x_p^i)^2, (1-x_p^i)^2\}]$, $[0, \sum_{p\in\boldsymbol{o}_j\backslash\boldsymbol{o}_i} \max\{(x_p^j)^2, (1-x_p^j)^2\}]$, and $[0, d - |\boldsymbol{o}_i \cup \boldsymbol{o}_j|]$, respectively. By computing the range of each entry $K_{i,j} = (K_o)_{i,j} \cdot (K_\Delta)_{i,j}$, we can determine the feasible domain of the optimization variable $K_\Delta$, denoted as $\boldsymbol{B}_l \preceq \boldsymbol{K}_\Delta \preceq \boldsymbol{B}_u$. In Stage II and the experimental sections, we will continue to use the Gaussian kernel as an example. However, it is important to note that our algorithm is applicable to various kernel functions, and we have provided several examples in Appendix A.

By leveraging the supervision information of the labels and performing alternating optimization between the kernel matrix and the classifier, we will obtain a kernel matrix that yields better classification performance. However, this flexible approach carries the risk of overfitting. To address this issue, our approach relies on training a robust classifier using optimization principles. Our objective is to ensure that the classifier performs well not only on $\tilde{K}$ but also on all possible outcomes within a norm sphere surrounding $\tilde{K}$. To achieve this, we introduce a perturbation variable denoted as $\mathcal{E}$ in the space of $\mathbb{R}^{N\times N}$. This variable serves as a mechanism to mitigate the potential impact of the aforementioned issue. Finally, we formulate the following optimization problem for Stage I:

$$
\begin{aligned}
\min_{\boldsymbol{K}_\Delta} \max_{\boldsymbol{\alpha},\mathcal{E}} \quad & \boldsymbol{1}^\top\boldsymbol{\alpha} - \frac{1}{2}\boldsymbol{\alpha}^\top\mathrm{diag}(\boldsymbol{y})\left(\boldsymbol{K}_o \odot \boldsymbol{K}_\Delta \odot \mathcal{E}\right)\mathrm{diag}(\boldsymbol{y})\boldsymbol{\alpha} + \eta\|\boldsymbol{K}_\Delta - \boldsymbol{1}\boldsymbol{1}^\top\|_{\mathrm{F}}^2 \\
\text{s.t.} \quad & \boldsymbol{B}_l \preceq \boldsymbol{K}_\Delta \preceq \boldsymbol{B}_u, \ \boldsymbol{K}_\Delta \in \mathcal{S}_+, \\
& \|\mathcal{E} - \boldsymbol{1}\boldsymbol{1}^\top\|_{\mathrm{F}}^2 \leq r^2, \ \mathcal{E} \in \mathcal{S}_+, \\
& \boldsymbol{y}^\top\boldsymbol{\alpha} = 0, \ \boldsymbol{0} \leq \boldsymbol{\alpha} \leq C\boldsymbol{1},
\end{aligned}
\tag{4}
$$

where the regularization parameters $\eta$ and $r$ are introduced to control the range of modification for the observed kernel matrix $\boldsymbol{K}_o$ and the intensity of the perturbation, respectively. By imposing PSD constraints on the variables $\boldsymbol{K}_\Delta$ and $\mathcal{E}$, it is ensured that the optimized kernel matrix remains PSD. As the value of $\eta$ increases, $\boldsymbol{K}_\Delta$ tends to approach the all-one matrix. In this scenario, the model approximates to a state where the missing values are filled with zeros. To solve the optimization problem outlined in (4), we propose an alternating optimization algorithm consisting of three steps.

**Step 1.** Optimizing $\boldsymbol{K}_\Delta$ with fixed $\mathcal{E}$ and $\boldsymbol{\alpha}$. In this step, problem in (4) with respect to $\boldsymbol{K}_\Delta$ is equivalent to

$$
\begin{aligned}
\min_{\boldsymbol{K}_\Delta} \quad & -\frac{1}{2}\boldsymbol{\alpha}^\top\mathrm{diag}(\boldsymbol{y})\left(\boldsymbol{K}_o \odot \boldsymbol{K}_\Delta \odot \mathcal{E}\right)\mathrm{diag}(\boldsymbol{y})\boldsymbol{\alpha} + \eta\|\boldsymbol{K}_\Delta - \boldsymbol{1}\boldsymbol{1}^\top\|_{\mathrm{F}}^2 \\
\text{s.t.} \quad & \boldsymbol{B}_l \preceq \boldsymbol{K}_\Delta \preceq \boldsymbol{B}_u, \ \boldsymbol{K}_\Delta \in \mathcal{S}_+.
\end{aligned}
\tag{5}
$$

The problem above is a semi-definite program, which is generally computationally expensive to solve. A common approach is to initially ignore the PSD constraint and solve the problem, followed by projecting the solution onto the space of PSD matrices (Cai et al., 2010; Liu et al., 2019). Defining $\Gamma_1 := \mathrm{diag}(\boldsymbol{\alpha} \odot \boldsymbol{y})\left(\boldsymbol{K}_k \odot \mathcal{E}\right)\mathrm{diag}(\boldsymbol{\alpha} \odot \boldsymbol{y})$, we have the final approximate solution for (5):

$$
\boldsymbol{K}_\Delta^* = \mathcal{P}_+\left(\mathcal{C}_{[\boldsymbol{B}_l, \boldsymbol{B}_u]}\left(\boldsymbol{1}\boldsymbol{1}^\top + \frac{1}{4\eta}\Gamma_1\right)\right).
\tag{6}
$$

The operator $\mathcal{C}_{[\boldsymbol{B}_l, \boldsymbol{B}_u]}(\boldsymbol{X})$ clips each element of $\boldsymbol{X}$ to the range defined by $[(\boldsymbol{B}_l)_{i,j}, (\boldsymbol{B}_u)_{i,j}]$. The operator $\mathcal{P}_+(\boldsymbol{X})$ performs eigenvalue decomposition on $\boldsymbol{X} = \boldsymbol{U}\boldsymbol{\Sigma}\boldsymbol{U}^\top$ and set any negative eigenvalues to 0, resulting in the new matrix $\hat{\boldsymbol{X}} = \boldsymbol{U}\boldsymbol{\Sigma}_+\boldsymbol{U}^\top$.

**Step 2.** Optimizing $\mathcal{E}$ with fixed $\boldsymbol{K}_\Delta$ and $\boldsymbol{\alpha}$. We will first solve the following problem:

$$\min_{\mathcal{E}} \quad \frac{1}{2}\boldsymbol{\alpha}^\top \mathrm{diag}(\boldsymbol{y}) \left(\boldsymbol{K}_k \odot \boldsymbol{K}_\Delta \odot \mathcal{E}\right) \mathrm{diag}(\boldsymbol{y})\boldsymbol{\alpha}$$
$$\text{s.t.} \quad \|\mathcal{E} - \mathbf{1}\mathbf{1}^\top\|_\mathrm{F}^2 \leq r^2. \tag{7}$$

Defining the Lagrange function $\mathcal{L}(\mathcal{E}, \lambda) := \frac{1}{2}\boldsymbol{\alpha}^\top \mathrm{diag}(\boldsymbol{y}) \left(\boldsymbol{K}_k \odot \boldsymbol{K}_\Delta \odot \mathcal{E}\right) \mathrm{diag}(\boldsymbol{y})\boldsymbol{\alpha} + \lambda \left(\|\mathcal{E} - \mathbf{1}\mathbf{1}^\top\|_\mathrm{F}^2 - r^2\right)$ by introducing the Lagrangian multiplier $\lambda$, we take the partial derivatives with respect to each of the two variables:

$$\frac{\partial \mathcal{L}}{\partial \mathcal{E}} = \frac{1}{2}\Gamma_2 + 2\lambda(\mathcal{E} - \mathbf{1}\mathbf{1}^\top) = 0, \tag{8a}$$

$$\frac{\partial \mathcal{L}}{\partial \lambda} = \|\mathcal{E} - \mathbf{1}\mathbf{1}^\top\|_\mathrm{F}^2 - r^2 = 0, \tag{8b}$$

where $\Gamma_2 := \mathrm{diag}(\boldsymbol{\alpha} \odot \boldsymbol{y}) \left(\boldsymbol{K}_k \odot \boldsymbol{K}_\Delta\right) \mathrm{diag}(\boldsymbol{\alpha} \odot \boldsymbol{y})$. From (8a), we can deduce that $\mathcal{E} - \mathbf{1}\mathbf{1}^\top = -\frac{1}{4\lambda}\Gamma_2$. Substituting it into (8b) yields $\frac{1}{4\lambda} = \frac{r}{\|\Gamma_2\|_\mathrm{F}}$. Therefore, the optimal solution for (7) is $\hat{\mathcal{E}} = \mathbf{1}\mathbf{1}^\top - \frac{r}{\|\Gamma_2\|_\mathrm{F}}\Gamma_2$. Subsequently, we project this solution onto the PSD matrix space to obtain the final solution for this step

$$\mathcal{E}^* = \mathcal{P}_+ \left(\mathbf{1}\mathbf{1}^\top - \frac{r}{\|\Gamma_2\|_\mathrm{F}}\Gamma_2\right). \tag{9}$$

**Step 3.** Optimizing $\boldsymbol{\alpha}$ with fixed $\boldsymbol{K}_\Delta$ and $\mathcal{E}$. Given $\boldsymbol{K}_\Delta$ and $\mathcal{E}$, the optimization problem is reduced to the standard SVM problem, which can be solved using various methods. In this case, we employ the gradient descent method to update $\boldsymbol{\alpha}$ in each iteration, and utilize the Adam optimizer (Kingma & Ba, 2014) to dynamically adjust the learning rate. After obtaining the updated variable $\hat{\boldsymbol{\alpha}}$, we proceed to project it onto the feasible set. This is done by first clipping it to the range $[0, C]$ and then calculating $\boldsymbol{\alpha}^* = \hat{\boldsymbol{\alpha}} - \frac{\boldsymbol{y}^\top \hat{\boldsymbol{\alpha}}}{N}\boldsymbol{y}$ as the final solution for this step.

### 3.3 STAGE II: DATA IMPUTATION WITH THE GIVEN MATRIX

In this stage, we will utilize the matrix $\tilde{\boldsymbol{K}}$ obtained from the previous training step to perform data imputation. We redefine the incomplete data set $\mathcal{D} = \{\boldsymbol{x}_{\boldsymbol{o}_i}^i, y_i\}_{i=1}^N$ as $\mathcal{D} = \{\boldsymbol{X}_o, \boldsymbol{y}, \boldsymbol{O}\}$, where $\boldsymbol{X}_o := [\boldsymbol{x}_{\boldsymbol{o}_1}^1 \; \boldsymbol{x}_{\boldsymbol{o}_2}^2 \; \ldots \; \boldsymbol{x}_{\boldsymbol{o}_N}^N] \in \mathbb{R}^{d \times N}$, $\boldsymbol{y} \in \mathbb{R}^N$ and $\boldsymbol{O} \in \{0, 1\}^{d \times N}$ is used to indicate which features are missing (represented by 0) and which features are observed (represented by 1). Next, we will compute $\Delta\boldsymbol{X} := [\Delta\boldsymbol{x}_1 \; \Delta\boldsymbol{x}_2 \; \ldots \; \Delta\boldsymbol{x}_N] \in \mathbb{R}^{d \times N}$ by using the entries of the trained kernel matrix $\tilde{K}_{i,j}$ as supervisory information. The goal is to minimize the discrepancy between the optimized kernel matrix calculated from the imputed data $\tilde{\boldsymbol{X}} = \boldsymbol{X}_o + \Delta\boldsymbol{X}$ and the original kernel matrix $\tilde{\boldsymbol{K}}$ through regression, i.e,

$$\min_{\{\Delta\boldsymbol{x}_i\}_{i=1}^N} \quad \sum_i \sum_j \left[\exp\left(-\gamma \left\|\boldsymbol{x}_{\boldsymbol{o}_i}^i - \boldsymbol{x}_{\boldsymbol{o}_j}^j + \Delta\boldsymbol{x}_i - \Delta\boldsymbol{x}_j\right\|_2^2\right) - \tilde{K}_{i,j}\right]^2.$$

By equivalently replacing the objective function and imposing location and range constraints on the imputation results, we obtain the following optimization problem for Stage II:

$$\min_{\Delta\boldsymbol{X}} \quad \sum_i \sum_j \left[\left\|\boldsymbol{x}_{\boldsymbol{o}_i}^i - \boldsymbol{x}_{\boldsymbol{o}_j}^j + \Delta\boldsymbol{x}_i - \Delta\boldsymbol{x}_j\right\|_2^2 + \frac{1}{\gamma}\ln\left(\tilde{K}_{i,j}\right)\right]^2$$
$$\text{s.t.} \quad \Delta\boldsymbol{X} \odot \boldsymbol{O} = \mathbf{0},$$
$$\mathbf{0} \preceq \boldsymbol{X}_o + \Delta\boldsymbol{X} \preceq \mathbf{1}. \tag{10}$$

The essence of this task involves solving a nonlinear system of equations. Additionally, by defining the missing ratio of data features as $m$, the aforementioned system consists of $N(N-1)/2$ equations

---

**Algorithm 1** Two-Stage Data Imputation Based on Support Vector Machine

---

**Input:** the incomplete data set $\mathcal{D} = \{\boldsymbol{X}_o, \boldsymbol{y}, \boldsymbol{O}\}$, the parameters $C, \gamma, \eta$, and $r$.
**Output:** the imputed data $\tilde{\boldsymbol{X}}$ and the combination coefficient $\boldsymbol{\alpha}$.
 1: **Stage I:**
 2: Compute $\boldsymbol{K}_o$, $\boldsymbol{B}_l$, and $\boldsymbol{B}_u$. Initialize $\boldsymbol{K}_\Delta^{(0)} = \mathcal{E}^{(0)} = \boldsymbol{1}\boldsymbol{1}^\top$ and $\boldsymbol{\alpha}^{(0)} = \frac{C}{2}\boldsymbol{1}$.
 3: **repeat**
 4:     Update $\boldsymbol{K}_\Delta^{(t_1)}$ by (6).
 5:     Update $\mathcal{E}^{(t_1)}$ by (9).
 6:     Update $\boldsymbol{\alpha}$ using gradient descent and project it onto the feasible set.
 7:     $t_1 = t_1 + 1$.
 8: **until** the stop criteria is satisfied.
 9: Compute $\tilde{\boldsymbol{K}} = \boldsymbol{K}_o \odot \boldsymbol{K}_\Delta$.
10: **Stage II:**
11: Initialize $\Delta\boldsymbol{x}_i^{(0)} = \frac{1}{2}\boldsymbol{1}$.
12: **repeat**
13:     **for** $i = 1$ to $N$ **do**
14:         Fix $\{\Delta\boldsymbol{x}_j^{(t_2-1)} : j \neq i\}$ and update $\Delta\boldsymbol{x}_i^{(t_2)}$ in (10).
15:     **end for**
16:     $t_2 = t_2 + 1$.
17: **until** the stop criteria is satisfied.

---

and $Ndm$ variables. Although the objective function in (10) is non-convex with respect to $\Delta\boldsymbol{X}$, the entries $\{K_{i,j}\}_{i,j=1}^N$ still provide abundant supervision information for the data imputation process in scenarios where the number of observed features ($d$) is significantly smaller than the total number of data ($N$) and the missing ratio ($m$) ranges between 0 and 1. To address this, we employ the BCD method to solve the optimization problem mentioned above. In each iteration, we select a specific column from $\Delta\boldsymbol{X}$ as a variable while keeping the remaining columns constant. We update the variable iteratively until reaching convergence. The accuracy of this solution is further validated by the experimental results presented later in this paper. The complete two-stage data imputation framework is summarized in Algorithm 1.

## 4 NUMERICAL EXPERIMENTS

### 4.1 EXPERIMENTAL SETTINGS

**Data sets and preprocessing.** We chose four real-world data sets from libsvm (Chang & Lin, 2011), namely *australian*, *german*, *heart* and *pima (a.k.a. diabetes)*. The details of these data sets are shown in Table 1. For preprocessing, we scaled $\boldsymbol{x}_i$ to $[0, 1]$ and $y_i$ to $\{-1, 1\}$. The data sets were then divided into three subsets: a training set, a complete holdout set for parameter selection, and a complete test set for evaluating and comparing the final results of the algorithms. The split was performed in a 4:3:3 ratio. For the training set, we constructed the missing data by randomly removing a given percentage of the features, and we defined the missing ratio of a data set as

Table 1: Data set statistics: $d$ and $N$ denote the number of data dimensions and the total number of data.

| Data Sets | $d$ | $N$ |
|-----------|-----|------|
| Australian | 14 | 690 |
| German | 21 | 1000 |
| Heart | 13 | 270 |
| Pima | 8 | 768 |

$$m := \frac{\#\text{ The missing features}}{\#\text{ The total features}}.$$

**Compared methods and parameters settings.** We selected a basic method along with three advanced methods for comparison with our proposed framework. To quantify the performance of the individual methods, we compared their classification accuracy on the test data set. All experiments were repeated 10 times, and the average accuracy of each method was reported. The implementation was carried out using MATLAB on a machine with an Intel® Core™ i7-11700KF CPU (3.60 GHz) and 32GB RAM. The source code will be released.

- **MI**: The missing values are imputed by setting them to the mean value of the corresponding features across all the available data, including both the training and test sets. We chose the penalty parameter $C \in \{2^{-5}, 2^{-4}, \ldots, 2^5\}$ and the bandwidth in the Gaussian kernel $\gamma \in \{2^{-5}, 2^{-4}, \ldots, 2^5\}$ using the holdout set.

- **GEOM** (Chechik et al., 2008): An iterative framework for optimizing a non-convex problem was introduced in this work. The main objective is to maximize the margin within the relevant subspace of the observed data. The parameters $C$ and $\gamma$ are chosen the same as in MI, and the iteration round $t$ is chosen from the set $\{2, 3, 4, 5\}$.

- **KARMA** (Hazan et al., 2015): This method proposed a new kernel function for the missing data $k_\beta(\cdot, \cdot) : (\mathbb{R} \cup \{*\})^d \times (\mathbb{R} \cup \{*\})^d \to \mathbb{R}$. We chose $C \in \{2^{-5}, 2^{-4}, \ldots, 2^5\}$ and $\beta \in \{1, 2, 3, 4\}$ using the holdout set.

- **genRBF** (Śmieja et al., 2019): It derived an analytical formula for the expected value of the radial basis function kernel over all possible imputations. The parameters $C$ and $\gamma$ were chosen the same as in MI.

- **Ours**: To avoid excessive parameter tuning, we utilized the parameters $C_{\text{MI}}$ and $\gamma_{\text{MI}}$ from MI and further selected $C \in \{C_{\text{MI}} \cdot 2^i, \ i = -1, 0, 1\}$ and $\gamma \in \{\gamma_{\text{MI}} \cdot 2^i, \ i = -1, 0, 1\}$. Additionally, we empirically set the values of $\eta = \|\boldsymbol{\alpha}_{\text{MI}}\|_2$ and $r = 0.2Nm$.

## 4.2 EXPERIMENTAL RESULTS

### 4.2.1 RESULTS OF DATA IMPUTATION FROM GIVEN KERNEL MATRIX

In order to ensure the operation of the entire framework, we first evaluated the performance of Stage II. We conducted experiments on the *heart* data set, considering various bandwidths of the Gaussian kernel and different missing ratios. We computed the complete kernel matrix $\boldsymbol{K}_{\text{gt}}$ based on the complete data $\boldsymbol{X}_{\text{gt}}$ and the chosen bandwidth

Table 2: Results of data imputation performance using a given kernel matrix with different Gaussian kernel bandwidths and missing ratios.

| Parameter | $m$ | $(e_{\boldsymbol{X}})_{\max}$ | $(e_{\boldsymbol{X}})_{\text{mean}}$ | $(e_{\boldsymbol{K}})_{\max}$ | $(e_{\boldsymbol{K}})_{\text{mean}}$ |
|---|---|---|---|---|---|
| $\gamma = 1$ | 10% | 3.46e−4 | 3.00e−7 | 5.95e−5 | 8.85e−8 |
| | 90% | 1.27e−1 | 6.71e−4 | 5.57e−2 | 8.28e−5 |
| $\gamma = \frac{1}{32}$ | 10% | 2.75e−4 | 2.88e−7 | 1.32e−5 | 4.52e−8 |
| | 90% | 1.52e−1 | 1.56e−3 | 3.70e−3 | 5.83e−5 |
| $\gamma = 32$ | 10% | 7.30e−4 | 4.74e−7 | 3.08e−5 | 6.81e−9 |
| | 90% | 3.81e−1 | 3.75e−3 | 1.03e−1 | 2.44e−5 |

$\gamma$. Subsequently, we randomly removed $Ndm$ features from the complete data, resulting in $\boldsymbol{X}_{\text{miss}}$. Next, utilizing the algorithm from Stage II in our framework, we performed data imputation on $\boldsymbol{X}_{\text{miss}}$ using the complete kernel matrix $\boldsymbol{K}_{\text{gt}}$, which yielded $\tilde{\boldsymbol{X}}$. In addition to caring about the quality of data imputation, we also focus on the quality of the kernel matrix used in predicting the class of new data. Therefore, we defined four evaluation metrics to assess the error between $\tilde{\boldsymbol{X}}$ and $\boldsymbol{X}_{\text{gt}}$, as well as the error between $\tilde{\boldsymbol{K}}$ and $\boldsymbol{K}_{\text{gt}}$: $(e_{\boldsymbol{X}})_{\max}$, $(e_{\boldsymbol{K}})_{\text{mean}}$, $(e_{\boldsymbol{K}})_{\max}$, and $(e_{\boldsymbol{K}})_{\text{mean}}$. We then conducted tests on our data imputation algorithm using different bandwidths of the Gaussian kernel and missing ratios. The results of these tests were reported in Table 2. We observed that at a low missing ratio ($m = 10\%$), both the imputed data and kernel matrix closely matched the ground truth. Even at a high missing ratio ($m = 90\%$), although there were some imputed features that deviated significantly from the true values, the overall performance of the algorithm in terms of average imputation remained highly accurate. Additionally, we found that when the Gaussian kernel parameter was particularly small ($\gamma = 1/32$) or large ($\gamma = 32$), the true kernel matrix tended to be a matrix of all ones or an identity matrix. In these scenarios, although there may be larger errors in the imputed data compared to when $\gamma = 1$, the kernel matrix used for actual predictions still maintained a good level of accuracy.

### 4.2.2 CLASSIFICATION RESULTS ON REAL-WORLD DATA SETS

Next, we compared different data imputation approaches using four real-world data sets. Since all imputation algorithms ultimately serve subsequent tasks, we evaluated their performance by calculating the mean and variance of their classification accuracy on the test data. The results were

Table 3: Comparison of classification accuracy (mean $\pm$ std) with a baseline and three state-of-the-art methods on four real data sets. We conducted testing on the scenarios with different missing ratios of the data. The best performance is highlighted in bold.

| Data Sets | Methods | $m$ | | | |
|---|---|---|---|---|---|
| | | 20% | 40% | 60% | 80% |
| Australian | MI | $0.866 \pm 0.022$ | $0.847 \pm 0.026$ | $0.825 \pm 0.028$ | $0.734 \pm 0.044$ |
| | GEOM | $0.849 \pm 0.025$ | $0.831 \pm 0.037$ | $0.754 \pm 0.050$ | $0.632 \pm 0.043$ |
| | KARMA | $0.861 \pm 0.020$ | $0.848 \pm 0.022$ | $0.831 \pm 0.045$ | $0.712 \pm 0.046$ |
| | genRBF | $0.867 \pm 0.016$ | $0.864 \pm 0.023$ | $0.796 \pm 0.041$ | $0.704 \pm 0.058$ |
| | Ours | $\mathbf{0.868 \pm 0.018}$ | $\mathbf{0.865 \pm 0.021}$ | $\mathbf{0.850 \pm 0.023}$ | $\mathbf{0.859 \pm 0.026}$ |
| German | MI | $0.722 \pm 0.023$ | $0.724 \pm 0.028$ | $0.715 \pm 0.022$ | $0.700 \pm 0.020$ |
| | GEOM | $0.723 \pm 0.028$ | $0.704 \pm 0.031$ | $0.695 \pm 0.024$ | $0.679 \pm 0.029$ |
| | KARMA | $0.731 \pm 0.019$ | $0.723 \pm 0.019$ | $0.706 \pm 0.036$ | $\mathbf{0.714 \pm 0.025}$ |
| | genRBF | $\mathbf{0.747 \pm 0.011}$ | $\mathbf{0.743 \pm 0.021}$ | $0.705 \pm 0.026$ | $0.687 \pm 0.045$ |
| | Ours | $0.721 \pm 0.026$ | $0.726 \pm 0.029$ | $\mathbf{0.722 \pm 0.023}$ | $0.706 \pm 0.030$ |
| Heart | MI | $0.816 \pm 0.031$ | $0.810 \pm 0.033$ | $0.782 \pm 0.044$ | $0.762 \pm 0.054$ |
| | GEOM | $0.806 \pm 0.021$ | $0.752 \pm 0.073$ | $0.758 \pm 0.053$ | $0.679 \pm 0.077$ |
| | KARMA | $0.784 \pm 0.040$ | $0.755 \pm 0.038$ | $0.756 \pm 0.031$ | $0.687 \pm 0.083$ |
| | genRBF | $0.813 \pm 0.031$ | $0.779 \pm 0.087$ | $0.737 \pm 0.032$ | $0.732 \pm 0.037$ |
| | Ours | $\mathbf{0.819 \pm 0.028}$ | $\mathbf{0.815 \pm 0.032}$ | $\mathbf{0.804 \pm 0.024}$ | $\mathbf{0.770 \pm 0.035}$ |
| Pima | MI | $0.765 \pm 0.023$ | $0.749 \pm 0.036$ | $0.728 \pm 0.028$ | $0.691 \pm 0.028$ |
| | GEOM | $0.721 \pm 0.031$ | $0.695 \pm 0.028$ | $0.680 \pm 0.058$ | $0.666 \pm 0.042$ |
| | KARMA | $0.747 \pm 0.034$ | $0.693 \pm 0.026$ | $0.668 \pm 0.034$ | $0.648 \pm 0.036$ |
| | genRBF | $\mathbf{0.781 \pm 0.011}$ | $0.755 \pm 0.017$ | $0.725 \pm 0.022$ | $0.659 \pm 0.029$ |
| | Ours | $0.751 \pm 0.016$ | $\mathbf{0.757 \pm 0.017}$ | $\mathbf{0.743 \pm 0.033}$ | $\mathbf{0.695 \pm 0.046}$ |

presented in Table 3. For situations with a relatively low missing data ratio, such as $m = 20\%$, the differences between the methods are not significant. Even using a simple method like MI can achieve decent predictive performance. When $m = 40\%$, our proposed method achieves the highest average accuracy and the lowest standard deviation on the *australian*, *heart*, and *pima* data sets, demonstrating the superior stability of our method. On the *german* data set, our algorithm's performance is second only to genRBF. When dealing with high missing data ratios, the prediction accuracy of the GEOM, KARMA, and genRBF methods fluctuates across different data sets. However, our method demonstrates even more significant advancements in such scenarios. For instance, at $m = 60\%$, our approach outperforms the second-ranking method by an additional accuracy improvement of approximately 0.02. This performance is further reflected in the case of $m = 80\%$, where our algorithm achieves precise and stable classification tasks. Additionally, detailed classification results for several data sets with inherent missing values can be found in Appendix B.

## 5 CONCLUSION

This paper proposed a novel two-stage data imputation framework, aiming to optimize the similarity relationships between data in order to guide the completion of missing features by pursuing better classification accuracy. In the first stage, we unify the tasks of kernel matrix imputation and classification within a single framework, enabling mutual guidance between the two tasks in an alternating optimization process to improve similarity relationships. The introduction of a perturbation variable enhances the robustness of the algorithm's predictions. In the second stage, we achieve, for the first time, the recovery of data features from a given kernel matrix, effectively utilizing the optimized information obtained in the first stage. By leveraging the supervision information through two stages, we have obtained a more flexible approach for data imputation, which provides significant advantages when dealing with high missing data rates. Numerical experiments have validated that our algorithm achieves higher prediction accuracy and demonstrates more stable performance compared to other methods on the test data.

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
