## A  LIST OF DECOMPOSED KERNEL FUNCTIONS

Below, we have provided a list of commonly used kernel functions that are decoupled. By separating the observed and unknown components, we compute the observed kernel matrix and establish an upper limit for the adjustment matrix. This approach allows us to fully leverage all the observed features in the data (assuming $\boldsymbol{x} \in [0, 1]^d$):

1. Linear Kernel:

$$K_{i,j} = \left(\boldsymbol{x}_{\boldsymbol{o}_i}^i\right)^\top \boldsymbol{x}_{\boldsymbol{o}_j}^j$$
$$= \sum_{p \in \boldsymbol{o}_i \cap \boldsymbol{o}_j} \left(x_p^i \cdot x_p^j\right) + \sum_{p \in \boldsymbol{o}_i \setminus \boldsymbol{o}_j} \left(x_p^i \cdot *\right) + \sum_{p \in \boldsymbol{o}_j \setminus \boldsymbol{o}_i} \left(* \cdot x_p^j\right) + \sum_{p \notin \boldsymbol{o}_i \cup \boldsymbol{o}_j} \left(* \cdot *\right)$$

2. Gaussian Kernel:

$$K_{i,j} = \exp\left(-\gamma D_{i,j}\right) \text{ where}$$
$$D_{i,j} = \|\boldsymbol{x}_{\boldsymbol{o}_i}^i - \boldsymbol{x}_{\boldsymbol{o}_j}^j\|_2^2$$
$$= \sum_{p \in \boldsymbol{o}_i \cap \boldsymbol{o}_j} \left(x_p^i - x_p^j\right)^2 + \sum_{p \in \boldsymbol{o}_i \setminus \boldsymbol{o}_j} \left(x_p^i - *\right)^2 + \sum_{p \in \boldsymbol{o}_j \setminus \boldsymbol{o}_i} \left(* - x_p^j\right)^2 + \sum_{p \notin \boldsymbol{o}_i \cup \boldsymbol{o}_j} \left(* - *\right)^2$$

3. Laplacian Kernel:

$$K_{i,j} = \exp\left(-\gamma D_{i,j}\right) \text{ where}$$
$$D_{i,j} = \|\boldsymbol{x}_{\boldsymbol{o}_i}^i - \boldsymbol{x}_{\boldsymbol{o}_j}^j\|_1$$
$$= \sum_{p \in \boldsymbol{o}_i \cap \boldsymbol{o}_j} |x_p^i - x_p^j| + \sum_{p \in \boldsymbol{o}_i \setminus \boldsymbol{o}_j} |x_p^i - *| + \sum_{p \in \boldsymbol{o}_j \setminus \boldsymbol{o}_i} |* - x_p^j| + \sum_{p \notin \boldsymbol{o}_i \cup \boldsymbol{o}_j} |* - *|$$

4. $\chi^2$ Kernel:

$$K_{i,j} = \sum_{p=1}^d \frac{2x_p^i x_p^j}{x_p^i + x_p^j}$$
$$= \sum_{p \in \boldsymbol{o}_i \cap \boldsymbol{o}_j} \frac{2x_p^i x_p^j}{x_p^i + x_p^j} + \sum_{p \in \boldsymbol{o}_i \setminus \boldsymbol{o}_j} \frac{2(x_p^i \cdot *)}{x_p^i + *} + \sum_{p \in \boldsymbol{o}_j \setminus \boldsymbol{o}_i} \frac{2(* \cdot x_p^j)}{* + x_p^j} + \sum_{p \notin \boldsymbol{o}_i \cup \boldsymbol{o}_j} \frac{2(* \cdot *)}{* + *}$$

Recall that $*$ represents an unknown real number within the range $[0, 1]$. The range of the second term depends on $x_p^i$, while the range of the third term depends on $x_p^j$. The range of the fourth term is $[0, d - |\boldsymbol{o}_i \cup \boldsymbol{o}_j|]$. Moreover, these kernel functions can all be adapted to the algorithm utilized in Stage II of our framework.

## B  EXPERIMENTAL RESULTS ON TWO DATA SETS WITH MISSING VALUES

In the experiments conducted in the main text, we primarily tested the algorithm's classification performance under the missing completely at random (MCAR) mechanism. In this section, we evaluated the performance of each method on two UCI data sets with missing values, namely *horse* and *cylinder*. In these two data sets, the missing values are not manually removed and the data collection process is not known to the user, so the missing mechanism may not be MCAR. Moreover, roughly speaking, the missing values tend to be more concentrated in a few features. And we selected NeuMiss (Le Morvan et al., 2020; 2021), a neural network-based completion method, as an additional comparison method. The details of these data sets are shown in Table 4 and the results are presented in Table 5.

Table 4: Data set statistics: $d$, $N$, and $m$ denote the number of data dimensions, the total number of data, and the missing ratio of a data set, respectively.

| Data Sets | $d$ | $N$ | $m$ |
|---|---|---|---|
| Horse | 25 | 368 | $\approx 30\%$ |
| Cylinder | 35 | 512 | $\approx 10\%$ |

Table 5: Comparison of classification accuracy (mean $\pm$ std) with a baseline and four state-of-the-art methods on two real data sets with missing values. The best performance is highlighted in bold.

| Data Sets | Methods | Accuracy |
|---|---|---|
| Horse | MI | $0.844 \pm 0.027$ |
| | GEOM | $0.848 \pm 0.021$ |
| | KARMA | $0.842 \pm 0.020$ |
| | genRBF | $0.836 \pm 0.028$ |
| | NeuMiss | $0.843 \pm 0.027$ |
| | Ours | $\mathbf{0.862 \pm 0.029}$ |
| Cylinder | MI | $0.648 \pm 0.014$ |
| | GEOM | $0.620 \pm 0.021$ |
| | KARMA | $\mathbf{0.678 \pm 0.018}$ |
| | genRBF | $0.620 \pm 0.021$ |
| | NeuMiss | $0.625 \pm 0.051$ |
| | Ours | $0.672 \pm 0.020$ |

On these two data sets, our algorithm also demonstrates excellent classification performance. On the *horse* data set, our accuracy is significantly better than other methods according to the paired t-test at the $5\%$ significance level. For the *cylinder* data set, predicting for this data set is challenging for all algorithms, and our algorithm performs slightly lower than KARMA but significantly higher than other methods. Furthermore, NeuMiss, though a well-designed neural network-based method, exhibits unsatisfactory performance on small data sets. This further highlights the significance of the kernel-based method we proposed.