# OpenReview forum: "Data Imputation by Pursuing Better Classification: A Supervised Learning Approach"
_ICLR.cc/2024/Conference — Submitted to ICLR 2024_

### Official Review · Reviewer_UxXs · 2023-10-31

**Soundness:** 3 good
**Presentation:** 3 good
**Contribution:** 3 good
**Rating:** 8
**Confidence:** 4

**Summary:**

The article addresses data imputation in a supervised classification setting. More precisely, the proposal frames the contribution within kernel-based approaches. By using weak assumptions on the similarity between the instances, the Gram matrix can be estimated so that the resulting classifier performs well with respect to the replacements made. Then, missing values can be identified using the obtained Gram matrix. The paper first presents the context and introduces the setting. Then, related works are briefly presented. The two-stage data imputation strategy is then detailed, before experiments are reported to study the properties of the proposal, and a short conclusion is drawn.

**Strengths:**

The paper is overall well written, in a clear and understandable way. The proposal is well described.

The contribution is sound.

**Weaknesses:**

Some key points (in particular, how parameters can be set) are not addressed.

Although technically sound and rational, the proposal could benefit from a deeper theoretical justification.

The results only mildly support the claim that the approach is superior to the others: they are often (but not always) better, but by a small margin only (and the difference cannot be deemed significant), and they are reported over four datasets only.

**Questions:**

General comments and questions:

The presentation of existing works is somehow a bit short: many works on learning from missing data or classification from missing data were not mentioned. There is no discussion on the nature of the missingness process here. The data seem to be considered as missing at random.

It seems that the approach detailed in Section 3.2 somehow corresponds to an adversarial optimization of the classifier (which should perform well "on all possible outcomes within a norm sphere surrounding $\tilde{\mathbf{K}}$"): can you elaborate on that ?

The remark regarding the results obtained for "extreme" values of $\gamma$ seem somehow obvious: the cases covered either correspond to instances being all dissimilar ($\gamma=1/32$), or similar ($\gamma=32$), hence the results. This raises the question of the sensitivity of your approach to the choice of $\gamma$—or, more generally, to the model parameters, the choice of which appears to be crucial, and actually very difficult to make without strong assumptions. Could you elaborate on that ?

The results do not seem to be significantly different between the various imputation approaches compared, the only exception being the Australian dataset, with $m=80\%$. Do you have any insights regarding this ?


More minor comments and questions:

It is not clearly stated whether the missing values are in the training or test data (or both).

Does ignoring the PSD constraint to solve the problem and then projecting back onto the space of PSD matrices have an impact on the result obtained ?

Should Step 2 be skipped in the first phase, what would be the outcome of the proposed strategy ? It seems that this amounts to implicitly assume that the data are "perfect": can you provide any insights regarding this ?

I do not understand why $\varepsilon^*$ should be positive definite (phase I, step 2, Equation (9), page 6). For complete consistency with Stage I, shouldn't $\varepsilon$ (or $\varepsilon^*$) be used as well in Stage II ?


Some suggestions on writing:

- page 1, "we typically use subsets of indices [...] with $\bfseries{x}_{\bfseries{o}_i}^i$": clearly define these notations;
- page 1, "the importance of labels has not been fully taken into account": this statement, somehow a bit assertive, is difficult to understand;
- page 2, "since there is $N-1$ supervising information available for each data": can you clarify ?
- page 4, Section 3.1, paragraph "Notations": sentences should not begin with a mathematical symbol;
- page 4, Section 3.2: matrix $\mathbf{K}_\Delta$ is not properly introduced;
- page 5, "is a semi-definite programming": should be "is a semi-definite program";
- page 6, $m$ does not seem to be properly introduced;
- page 7, "including \textit{australian}, [...]": should be "namely \textit{australian}, [...]".

---

> ### Author Response · Authors · 2023-11-17
>
> Thank you for your recognition and detailed feedback on our work. We have addressed your concerns as follows.
>
> **R4.1 The settings and the choice of the model parameters.**
>
> Thank you for pointing out this issue in detail. It is indeed a meaningful consideration for users. We empirically offer some suggestions regarding the parameter settings. Regarding the adjustment of the parameter $\gamma$, we adopt the traditional "grid search" approach commonly used in SVM to select the parameter value.  For $\eta$, which controls the extent to which the similarity between samples can be modified, if there is still room for improvement in the predictive accuracy of the training set, reducing $\eta$ can be explored to seek better similarity relationships among samples. As for $r$, which governs the robustness of the classifier, if the predictive accuracy during training is already high but the performance on the holdout set is unsatisfactory, increasing the value of $r$ can help alleviate this issue. We wish that the above response partially addresses your concerns.
>
> **R4.2 Studying on more data sets.**
>
> We sincerely appreciate your valuable feedback. Following your suggestion, we conducted experiments on two new data sets. For detailed results of the experiments, please refer to **R0.1** in the Author Rebuttal.
>
> **R4.3 Studying on other works and other missingness processes.**
>
> We sincerely appreciate your insightful comments. Based on your suggestions, we have included a comparison of results on two data sets with inherent missing values. And we compare the results with a recent neural network-based method, NeuMiss [1]. For more details, please refer to **R0.1** in the Author Rebuttal.
>
> **R4.4 Elaboration of the adversarial optimization of the classifier.**
>
> In Section 3.2, our approach is primarily inspired by the principles of robust optimization, which shares similarities with the adversarial optimization you mentioned. Both approaches aim to ensure good performance of the model even under worst-case scenarios.
>
> **R4.5 The difference among the results of different methods.**
>
> We have indeed observed the issue you pointed out, and this phenomenon has also been observed in other studies. For example, Table 1 in [2] and Table 1(c) in [3] demonstrate that there is no significant difference in test accuracy between MI, 0-imputation, and the proposed method in certain scenarios. We think that these observations may be attributed to several factors, including the proportion of missing data and intrinsic characteristics of the data. Firstly, for simple cases where the proportion of missing data is low, as shown in Table 3 with the example of Australian $(m=20\%)$, all the methods performed well. In more challenging scenarios, data imputation becomes exceedingly difficult, causing each method to struggle in achieving improved results, as seen in Table 3 with the example of German under high missingness.
>
> **R4.6 Missing values: Presence in training/test sets or not.**
>
> In our experimental setup, the training set contains missing values while the test set does not, which follow the settings in [4].
>
> **R4.7 The influence of dealing with the PSD constraint.**
>
> Such an operation does not affect the results and is a widely applicable operation. For detailed theoretical references, please refer to the Theorem 2.1 in [5] and the Equation (9) in [6].
>
> **R4.8 The outcome when skipping the Step 2 in the first phase.**
>
> If we skip Step 2, the flexibility of our framework (allowing changes in both data similarity and classifier) makes it susceptible to overfitting. In such a scenario, if the allowed range for $K_\Delta$ to change is large (i.e., when the value of $\eta$ is set too small), it can lead to the situation you mentioned as "perfect data"，where the resulting kernel matrix tends to approach the ideal kernel matrix $yy^T$ as closely as possible. Although it results in high classification accuracy on the training set, it may not generalize well to the test set.
>
> **R4.9 Discussions about the $\mathcal{E}^\ast$.**
>
> a) We set $\mathcal{E}^\ast$ as positive semi-definite (PSD) to ensure that the kernel matrix used in SVM remains PSD, as supported by the Schur Product Theorem [7].
>
> b) It is important to note that that we pursue the similarity relationship between data with better classification performance by optimizing $K_\Delta$, and use these relationships to guide the data imputation process. The introduction of $\mathcal{E}^\ast$ is solely intended to enhance the robustness of the classifier. Therefore, in Stage II, where we focus on data feature imputation, we only utilize $K_o\odot K_\Delta$.

---

> ### Author Response · Authors · 2023-11-17
>
> **R4.10 The modification on writing.**
>
> Thank you very much for your valuable suggestions and careful reading. In response to each of your suggestions, we have presented the modified sentences below.
>
> a) Page 1. we typically use the set of observable features $o_i\subseteq 2^{\lbrace 1,\cdots,d\rbrace }$ and missing features $\ast$ to represent the missing data $x_{o_i}^i\in(\mathbb{R}\cup \{\ast\})^d$. This sequence is then consolidated into an incomplete data set $\mathcal{D}=\lbrace x_{o_i}^i, y_i\rbrace _{i=1}^N$.
>
> b) Page 1. However, the effective utilization of label information has not been fully explored yet.
>
> c) Page 2. This step is relatively easy, particularly in the traditional scenario where $N \gg d$. Because for each data, there is $N – 1$ available supervised information stored in the kernel matrix to guide the imputation of $d$ or fewer elements.
>
> d) Page 4. The set of real numbers is written as $\mathbb{R}$. The set of integers from 1 to $N$ is written as $[N]$. The cardinality of the set $\mathcal{A}$ is written as $|\mathcal{A}|$ denotes. We take $a$, $\boldsymbol{a}$, and $\boldsymbol{A}$ to be a scalar, a vector, and a matrix, respectively. Let $\boldsymbol{0}$ and $\boldsymbol{1}$ denote vectors consisting of all zeros and all ones with the appropriate size. The inner product of two vectors in the given space is written as $\langle\cdot, \cdot\rangle$. We take $\mathrm{diag}(\boldsymbol{a})$ to be an operator that extends a vector to a diagonal matrix. The Frobenius norm of a matrix is written as $\lVert \cdot \rVert_\mathrm{F}$. The set of positive semi-definite (PSD) matrices is written as $\mathcal{S}_{+}$. The Hadamard product is written as $\odot$.
>
> e) Page 4. Initially, we compute the observed kernel matrix $K_o$, and then optimize an adjustment matrix $K_{\Delta}$ with the same dimension to derive $\tilde{K}=K_o\odot K_{\Delta}$ as the imputed kernel matrix.
>
> f) Page 5. The problem above is a semi-definite program, which is generally computationally expensive to solve.
>
> g) Page 6. The essence of this task involves solving a nonlinear system of equations. Additionally, by defining the missing ratio of data features as $m$, the aforementioned system consists of $N(N-1)/2$ equations and $Ndm$ variables.
>
> h) Page 7. We chose four real-world data sets from libsvm (Chang & Lin, 2011), namely australian, german, heart and pima (a.k.a. diabetes).
>
> **Referneces**
>
> [1] Le Morvan M, Josse J, Moreau T, et al. NeuMiss networks: differentiable programming for supervised learning with missing values[J]. Advances in Neural Information Processing Systems, 2020, 33: 5980-5990.
>
> [2] Chechik G, Heitz G, Elidan G, et al. Max-margin Classification of Data with Absent Features[J]. Journal of Machine Learning Research, 2008, 9(1).
>
> [3] Hazan E, Livni R, Mansour Y. Classification with low rank and missing data[C]//International conference on machine learning. PMLR, 2015: 257-266.
>
> [4] Pelckmans K, De Brabanter J, Suykens J.A.K, et al. Handling missing values in support vector machine classifiers[J]. Neural Networks, 2005, 18(5-6): 684-692.
>
> [5] Cai J F, Candès E J, Shen Z. A singular value thresholding algorithm for matrix completion[J]. SIAM Journal on optimization, 2010, 20(4): 1956-1982.
>
> [6] Liu F, Huang X, Gong C, et al. Learning data-adaptive non-parametric kernels[J]. The Journal of Machine Learning Research, 2020, 21(1): 8590-8628.
>
> [7] Styan G.P.H. Hadamard products and multivariate statistical analysis[J]. Linear algebra and its applications, 1973, 6: 217-240.

---

> ### Author Response · Authors · 2023-11-20
>
> Dear reviewer UxXs,
>
> We've taken your feedback on board and implemented the necessary revisions in the paper before submitting it to the OpenReview system. Here are a few of the main changes we made:
>
> * We've expanded the discussion on existing work in the related work section on page 4, and we've added a comparison with a recent neural network method called NeuMiss in Appendix B.
> * Appendix B also includes a discussion of the algorithm's results on non-missing completely at random data.
> * We have provided clearer explanations in the "Data sets and preprocessing" section in Section 4.1 regarding whether missing values exist in the training and test sets.
> * We have made the necessary modifications in the text to address all eight of your meticulous suggestions on writing.
>
> We sincerely appreciate your valuable suggestions and detailed feedback. They have helped enrich the content of our paper and make it more rigorous. We hope that the modifications we have made meet your standards. If you find any areas where our responses may still be inadequate or if you have any other specific concerns, we welcome your additional insights. We will make every effort to respond and engage in further discussion before the end of the discussion period.

---

> > ### Comment · Reviewer_UxXs · 2023-11-21
> > **Following authors' rebuttal**
> >
> > I would like to thank the authors for their detailed answers to my questions and comments. I believe it has improved the quality of the contribution, and I have consequently raised my score.

---

> > > ### Author Response · Authors · 2023-11-21
> > >
> > > Thank you for acknowledging the contribution of our method. We will incorporate your detailed comments into the final version of the manuscript. We sincerely appreciate the time and effort you dedicated to reviewing our manuscript.

---

### Official Review · Reviewer_VYgf · 2023-10-31

**Soundness:** 3 good
**Presentation:** 3 good
**Contribution:** 3 good
**Rating:** 8
**Confidence:** 4

**Summary:**

This paper proposes a two stage procedure for dealing with missing data targeted towards classification.  In the first stage, the authors propose a method which jointly finds a Kernel matrix K_{\Delta} and solves a dual SVM formulation.  Then, in the second stage, the authors use the Kernel matrix K_{\Delta} \odot K_o to perform data imputation via solving a non-convex optimization problem provided in Eq. (10).

**Strengths:**

Both stages of the proposed procedure are interesting and novel.  Moreover, the experimental results are promising.  The writing is also quite clear.

**Weaknesses:**

The main weakness lies in the choice of experiments and settings.  In particular, the authors consider a MCAR (missing completely at random) set up for the experiments in which they induce the missingness pattern in the data.  I do appreciate the results in Table 3 and the differentiation based on the amount of missing data, however, it would be stronger to include further validation.

**Questions:**

Would it be possible to include an experiment and compare the different methods on a dataset with missing data in which the missingness is not induced artificially? For instance, taking a dataset with missing entries but for which there are sufficiently many labels to train your method and the other methods to which you compare.  This would help to understand the method you propose beyond the artificial MCAR setting, which would be important.  I would be willing to raise my score if an experiment of this sort were to be included.

---

> ### Author Response · Authors · 2023-11-17
>
> Thank you very much for finding our two-stage procedure interesting. Indeed, both the optimization of pairwise data similarity relationships in the first stage and the subsequent data recovery using the kernel matrix are novel aspects that contribute to the effectiveness of our imputation task. We also appreciate your recognition of our comprehensive testing of algorithm performance under various data missing ratios. We have addressed your concerns as follows.
>
> **R3.1 Studying data sets beyond artificial MCAR settings.**
>
> We sincerely appreciate your highlighting the need for result validation in non-artificial MCAR settings. As you rightly pointed out, obtaining results under different missing mechanisms would provide stronger support for the superiority of our algorithm. Reviewer Jet1 also raised a similar question. Therefore, we have addressed this issue in **R0.1**, and please let us know if it is appropriate. If you have any further questions or if there are other specific aspects of the results that interest you, we would be more than happy to engage in further discussions with you.

---

> ### Author Response · Authors · 2023-11-20
>
> Dear reviewer VYgf,
>
> Following your suggestions, we've completed the revisions and uploaded the revised version of the paper to the OpenReview system. We have made the following additions to the paper:
>
> * We have addressed the algorithm comparison results beyond the artificial MCAR setting in Appendix B. Additionally, we have included a comparison with a recent neural network-based method.
>
> We greatly appreciate your suggestions, as they have helped enrich the content of our paper. We hope that the additional experiments we have included further validate the performance of our algorithm. If you would like to see how our algorithm performs in other scenarios or have any other specific concerns, we will do our best to respond and discuss them before the end of the discussion period.

---

> > ### Comment · Reviewer_VYgf · 2023-11-20
> > **Thank you and brief follow-up**
> >
> > Thank you to the authors for including this experiment in such a short amount of time.  I do think this improves the paper quite a bit, but I am not sure it warrants an increase in score.  One clarification which would help to understand is: Why did you select these two datasets and what happens if you select a larger dataset?
> >
> > Also, a small issue, but you say in appendix B that the missing values do not follow MCAR.  How do you know that they are not MCAR? Without evidence either way, it may be more appropriate to clarify that the missingness pattern may not be MCAR, since it was not induced artificially and the data collection process is not known to the user.

---

> > > ### Author Response · Authors · 2023-11-20
> > >
> > > Many thanks for your prompt response and acknowledgment of our improvements.
> > >
> > > As you mentioned, the time allocated for this discussion is indeed limited, making it challenging to expand experiments to include additional data sets. Consequently, we decided to concentrate on two widely used data sets with missing values, namely Horse and Cylinder. These data sets are commonly referenced in various works, such as [1-4]. It's worth noting that both data sets exhibit relatively high missing rates (30% and 10%, respectively) compared to other publicly available data sets with missing values. The elevated missing rates pose a greater challenge for data imputation, better highlighting the superior performance of our method.
> > >
> > > As discussed in **R0.2**, our method relies on pairwise information among the data, resulting in the management of an $N\times N$ kernel matrix. This poses challenges in terms of time and computational resources when dealing with large-sized datasets. While there are existing acceleration methods for handling large kernel matrices, our algorithm's kernel matrix is utilized for data imputation. This restricts the use of conventional approximation-based acceleration methods, such as Nystr&ouml;m methods or Random Fourier Features. Currently, there are no available acceleration techniques specifically tailored for kernel-based data imputation methods. Consequently, the development of new acceleration techniques becomes necessary if kernel-based methods are to be employed for large dataset data imputation. This direction is indeed one of the areas we plan to explore in our future research.
> > >
> > > Thank you for pointing out the imprecise expression regarding the dataset. We will rectify this description in the final version.
> > >
> > > We value your interest and insights into our paper and hope our explanation enhances your confidence in our work.
> > >
> > > **Reference**
> > >
> > > [1] Madhu, G., and G. Nagachandrika. "A new paradigm for development of data imputation approach for missing value estimation." _International Journal of Electrical and Computer Engineering_ 6, no. 6 (2016): 3222.
> > >
> > > [2] Dinh, Duy-Tai, Van-Nam Huynh, and Songsak Sriboonchitta. "Clustering mixed numerical and categorical data with missing values." _Information Sciences_ 571 (2021): 418-442.
> > >
> > > [3] Kunanbayev, Kassymzhomart, Islambek Temirbek, and Amin Zollanvari. "Complex encoding." In 2021 International Joint Conference on Neural Networks (IJCNN), pp. 1-6. IEEE, 2021.
> > >
> > > [4] Nugroho, Heru, Nugraha Priya Utama, and Kridanto Surendro. "Normalization and outlier removal in class center-based firefly algorithm for missing value imputation." _Journal of Big Data_ 8, no. 1 (2021): 1-18.

---

> > > > ### Comment · Reviewer_VYgf · 2023-11-21
> > > > **Follow up**
> > > >
> > > > Thanks very much for your quick response.  I think this is a nice contribution, and with these additional experiments, I am happy recommend acceptance and have increased my score to 8.

---

> > > > > ### Author Response · Authors · 2023-11-21
> > > > >
> > > > > We are delighted by your recognition of our work and sincerely appreciate your valuable comments and feedback throughout the review and discussion process. We will continue to improve our manuscript based on your and other reviewers' valuable insights.

---

### Official Review · Reviewer_h2QN · 2023-11-03

**Soundness:** 3 good
**Presentation:** 3 good
**Contribution:** 2 fair
**Rating:** 5
**Confidence:** 3

**Summary:**

The paper proposes a new kernel-based method for dealing with data missing at random. Their basic idea is to estimate them through the learned imputed kernel matrix. Their main novelty compared to previous papers is in making use of unknown features as well as observed ones while learning this kernel. Once the imputed kernel matrix is obtained, the missing values are estimated numerically by minimizing
minimize the discrepancy between the imputed kernel matrix calculated from the imputed data. The proposed method is compared with 4 other strong baselines on 4 benchmark data sets. The results indicate that the proposed approach could be better than the baselines particularly when the proportion of the missing data is large.

**Strengths:**

+ The paper is generally well written and easy to follow
+ The proposed approach is well justified and explained clearly enough
+ The experimental results indicate that the proposed approach might be better than the baselines

**Weaknesses:**

- This paper is mostly an incremental contribution compared to the state of the art
- There is a long history of research on imputation of data missing at random. Thus, comparing to only 4 other baselines on 4 small data sets (both in number of examples and features) might not be comprehensive enough. It is not clear why those particular 4 data sets were selected (other than being very small).
- Looking at the error bars, for most of the results the improvement does not seem to be statistically significant

**Questions:**

In addition to the previous comments, it would be useful to show the computational cost for the performed experiments. The largest data set used in the experiments has only 1000 examples. Is this because the proposed method is too expensive to run on larger data?

---

> ### Author Response · Authors · 2023-11-17
>
> Thank you for recognizing the novelty of our approach in leveraging both unknown and observed feature information when optimizing the kernel matrix and the justification of our approach. In fact, it is precisely by considering this pairwise information among the data that we achieve better imputation results and, consequently, improve the classification performance (especially in scenarios with high data missing rates). We address your concerns as follows.
>
> **R2.1 Studing on more data sets and imputation methods.**
>
> We sincerely appreciate your valuable feedback. The additional comparisons you mentioned regarding the datasets and imputation methods are indeed important for validating the performance of our algorithm, and other reviewers have also expressed similar concerns. Therefore, following your suggestion, we conducted experiments using the latest neural network-based method, NeiMiss [1], on two additional datasets. And we have provided detailed results in **R0.1** in the Author Rebuttal. We hope this addresses some of your concerns.
>
> **R2.2 Studing on larger data sets.**
>
> Regarding the issue of scalability to large-scale data in our proposed method, it is indeed a challenge due to the need for leveraging pairwise information in the samples. Additionally, some popular techniques used to accelerate kernel methods cannot currently be applied to the imputation problem. This issue will also be a research direction that we will dedicate efforts to in the future. For a more detailed discussion, please refer to **R0.2** in the Author Rebuttal. If you have any specific questions, we are more than happy to engage in further discussions with you over the next week.
>
> **Reference**
>
> [1] Le Morvan M, Josse J, Moreau T, et al. NeuMiss networks: differentiable programming for supervised learning with missing values[J]. Advances in Neural Information Processing Systems, 2020, 33: 5980-5990.

---

> ### Author Response · Authors · 2023-11-20
>
> Dear reviewer h2QN,
>
> We've made the changes you suggested in the main text and uploaded the revised paper to the OpenReview system. And we have listed the main changes below.
> * We added a discussion on some recent imputation methods in the related work section on page 4.
> * In Appendix B, we included a separate discussion on the algorithm results obtained from two datasets with inherent missing values. Additionally, we introduced and compared a recent neural network-based method to further highlight the advancements of our approach.
>
> We greatly appreciate your suggestions, which have helped enrich the content and improve the quality of our paper. In addition to the modifications made in the text, we have also addressed some discussions regarding the case of large scale data in our previous rebuttal. We hope that our responses have addressed your concerns. We would be more than happy to engage in further communication with you. If you have any other concerns or related questions, we will make every effort to respond before the end of the discussion period.

---

### Official Review · Reviewer_Jet1 · 2023-11-10

**Soundness:** 2 fair
**Presentation:** 2 fair
**Contribution:** 3 good
**Rating:** 3
**Confidence:** 3

**Summary:**

The authors tackle supervised learning with missing values, using support vector machines (SVMs).

To this end, they design a new way to learn a kernel matrix of an incomplete data set, by optimising the SVM loss (what is called "Stage I" in their algorithm). After this kernel has been learned, they use it to impute the data set by minimising the squared error between the kernel of the imputed data set and the learned kernel (this is "Stage 2").

They do several experiments on four real classification data sets.

**Strengths:**

The two stages of the algorithms both involve quite clever ideas.

The first stage in particular is based on the idea of treating the terms of the full kernel matrix that depend on incomplete data as parameters to be optimised. This is an excellent idea that is, to the best of my knowledge, novel.

The main idea behind the second stage is less innovative but very sensible. I also appreciate the fact that this "stage II" is empirically investigated on its own in Section 4.2.1.

----- POST-REBUTTAL EDIT -------

After the discussion with the authors, and giving this quite some thought, I have decided to change my score to 3 because of the maths clarification that I asked (point 3 of the "Weaknesses") that turned out to be a mistake that makes the algorithm potentially not valid (the constraints on the matrix may not be respected).

While the additional experiments were welcome, I do not think they were completely satisfactory, since the strongest baseline remains mean imputation (and, as discussed with the authors, one of the added baselines, Neumiss, is a linear regression method, which is not really fit for nonlinear classification). Adding stronger baselines would clearly make the experiments more compelling.

The final score is a bit harsh, and I reiterate that I believe the key ideas of the paper are quite good. I just don't think the paper is ready to be published yet. If the paper is accepted, I strongly encourage the authors to try to add a strong baseline to the tables (eg gradient boosting), to acknowledge that the claimed constraints might not be respected by the algorithm, and to clarify the data-preprocessing (normalisation is not an obvious task where there are missing data).

**Weaknesses:**

Main concerns

1) My main concern is related to the experiments, that have several issues, in my opinion.

a) Studying only 4 small data sets ($N\leq 1,000$) is not particularly compelling, especially given that most standard deviations are quite important (in Table 3, I doubt the author's technique is statistically significantly better than mean imputation in most scenarios).

b) Mean imputation appears to be, by far, the best method (if we ignore the author's method). This is not very consistent with the literature. For instance, in the genRBF paper (Smieja et al., 2019), genRBF is on par or better than the mean on the "Australian" data set, while it is much worse than the mean in this submission. Similarly, in the GEOM paper (Chechik et al., 2008), GEOM is essentially always on par with the mean, and is generally worse in this submission.

2) The authors do not study the theoretical properties of their methods. In particular, the assumptions on the missingness mechanism are not discussed. All experiments use missing completely at random (MCAR) data, and the authors do not discuss this experimental design choice. Studying (empirically and/or theoretically) whether or not this algorithm works on non-MCAR data would be interesting.

Secondary concerns

3) The paper read generally well, but the mathematics are at times quite unclear. Several objects are not properly defined and some facts are not really proven, for instance
- I imagine $K_0 = \exp ( - \gamma \sum_{p \in o_i \cap o_j} (x^i_p - x^j_p)^2 )$, but $K_0$ is never defined,
- in Equation (3), the mathematical meaning of $ (x^i_p - *) $ is unclear
- why is $K^*_\Delta$ in Equation (6) in the proper range (between $B_l$ and $B_u$)? After clipping, you project on positive semidefinite matrices, why is it guaranteed that it would be still in the right range?

Minor things

- I find it a bit odd to call $\mathcal{E}$ a "noise", since it not something random but something that you optimize
- I also find the phrase "imputed kernel matrix", used a few times (in different forms) a bit odd: this matrix is always a complete matrix, it is the dataset used to build it that needs to be imputed
- There has been a significant amount of work on supervised learning with missing values recently. Some of these papers could be interesting to discuss, for instance:

Josse et al., On the consistency of supervised learning with missing values, arXiv:1902.06931, 2020

Le Morvan et al., What’s a good imputation to predict with missing values? NeurIPS 2021

Bertsimas et al., Beyond Impute-Then-Regress: Adapting Prediction to Missing Data, arXiv:2104.03158, 2022

**Questions:**

- see questions in the "Weaknesses" section, point 3

- In your experiments, I did not understand if you used as a final classifier SVM with $\tilde{K}$ as a kernel matrix, or with the kernel matrix of the imputed data set ?

---

> ### Author Response · Authors · 2023-11-17
>
> Thanks for appreciating the novelty of our work and for providing insightful comments. We address your concerns as follows.
>
> **R1.1 Studing on more and larger data sets.**
>
> We sincerely appreciate your valuable feedback. Following your suggestion, we conducted experiments on two new data set. For detailed results of the experiments, please refer to **R0.1** in the Author Rebuttal. As for the lack of involvement with large-scale data sets, we provided a comprehensive discussion in **R0.2** in the Author Rebuttal.
>
> **R1.2 New experimental results for two tested items in Australian and one tested item in German.**
>
> Your questions about the accuracy reported for MI, genRBF, and GEOM may be related to our experimental setup. For specific examples, we will discuss them as follows.
>
> a) For MI, as described in [2], it allows leveraging feature information from the test set to aid in imputation. In our experimental settings, we only introduced missing features in the training set while utilizing complete holdout and test sets. Therefore, especially in cases of high feature missing rates, the information from the test set contributes to better classification performance for MI.
>
> b) For genRBF, some abnormal results in extreme cases were due to the narrow range used to adjust hyperparameters. We have modified the grid search range for $C$ and $\gamma$ and made our best efforts to improve the performance, and then we have adjusted the values of genRBF for several tested items. For the case of Australian $(m=60\%)$, Australian $(m=80\%)$, and German $(m=80\%)$, based on our correction, genRBF should have the accuracy of $0.796 \pm 0.041$, $0.704 \pm 0.058$, and $0.687 \pm 0.045$, respectively. And for other scenarios, we carefully examined and confirmed that the previously used parameters were optimal. Thank you for meticulously pointing out this issue and we will revise these results in our manuscript.
>
> c) For GEOM, we conducted experiments specifically for the data-missing scenario of $m=90\%$ (i.e., the experimental setup described in their paper), using the provided code by the authors. On the Pima dataset, we obtained a result of $0.658 \pm 0.029$ for the GEOM method, which aligns with the reported result of $0.66 \pm 0.05$ in the Table 1 of [2]. However, we obtained a result of $0.689 \pm 0.036$ for the MI, which is higher than the reported value of $0.65 \pm 0.04$ in the paper. This discrepancy arises from the utilization of the complete test set, as we mentioned above.
>
> **R1.3: Studing on the non-MCAR data.**
>
> We sincerely appreciate your valuable feedback. Based on your suggestions, we have included a comparison of results on two data sets with inherent missing values, which are non-MCAR. For more details, please refer to **R0.1** in the Author Rebuttal.
>
> **R1.4: Modifications in object definitions and the mathematics.**
>
> We sincerely appreciate your meticulousness in pointing out our oversights.
>
> a) Regarding the definition of $K_o$, as you correctly noted, it should indeed be defined by the first term in the last line of Equation (3).
>
> b) In regards to the symbol $\ast$, it represents an unknown real number in future manuscript, which aids in computing the range of values such as $x_p^i-\ast$, assuming that $x$ is normalized to the range [0,1].
>
> c) Thank you very much for your detailed observation. Indeed, our wording was not precise. Clipping a matrix first and then projecting it into the PSD matrix space does not guarantee that every element of the resulting matrix remains within the proper range. Our intention in introducing the proper range was merely to prevent significant changes in $K_\Delta$. While there may be elements slightly exceeding the range after the projection, these modified values can still be bounded, which allows the original constraint to remain effective and leads to similar performance in subsequent algorithms.
>
> We will make the necessary modifications and additions in the manuscript to address the issues you have pointed out in the manuscript.

---

> > ### Author Response · Authors · 2023-11-17
> >
> > **R1.5 Discussions about the ‘minor things’.**
> >
> > a) Thank you for your suggestion. We will revise it to "perturbation matrix" in the subsequent manuscript.
> >
> > b) Regarding the phrase "imputed kernel matrix", we will replace it with "optimized kernel matrix" in order to express the intended meaning more clearly.
> >
> > c) Thank you very much for providing these articles on handling missing values in supervised learning. We have included one of the methods, NeuMiss [3], in **R0.1** and **R0.2** for comparison with our approach. The mentioned articles are indeed very interesting, and we will incorporate and discuss these works in our future manuscript to further enhance its quality.
> >
> > **R1.6 The kernel matrix used in the final SVM.**
> >
> > $\tilde{K}$ is the kernel matrix used in the final SVM classifier.
> >
> > **Referneces**
> >
> > [1] Śmieja M, Struski Ł, Tabor J, et al. Generalized RBF kernel for incomplete data[J]. Knowledge-Based Systems, 2019, 173: 150-162.
> >
> > [2] Chechik G, Heitz G, Elidan G, et al. Max-margin Classification of Data with Absent Features[J]. Journal of Machine Learning Research, 2008, 9(1).
> >
> > [3] Le Morvan M, Josse J, Moreau T, et al. NeuMiss networks: differentiable programming for supervised learning with missing values[J]. Advances in Neural Information Processing Systems, 2020, 33: 5980-5990.

---

> ### Author Response · Authors · 2023-11-20
>
> Dear reviewer Jet1,
>
> We've uploaded the revised paper to the OpenReview system. Based on your suggestions, here's what we've done:
>
> * In Appendix B, we added experiments on non-MCAR data and included one of the recent works on supervised learning with missing values that you mentioned.
> * We checked the accuracy reported for genRBF, and addressed the issue of abnormal values in extreme cases by adjusting the range of hyperparameters in the grid search. Consequently, we made modifications to the results of the three items in Table 3 on page 9.
> * Under Equation 3 on page 5, we explained the meaning of the $\ast$ and added a description of $K_o$. We also corrected the name of $K_\Delta^\ast$ on page 5.
> * We replaced the terms $\mathcal{E}$ and "imputed kernel matrix" with "perturbation" and "optimized kernel matrix", respectively.
> * We added a discussion of the several pieces of work you mentioned to the last category of related work on page 4.
>
> We truly appreciate your valuable suggestions and detailed feedback. We hope that these modifications meet your standards and make our paper more rigorous and reader-friendly. We're also curious if you'll have the opportunity to review our rebuttal. If you have any other concerns or related questions, we would be delighted to discuss and communicate with you before the end of the discussion period.

---

> > ### Comment · Reviewer_Jet1 · 2023-11-22
> >
> > Many thanks for the thorough revision of the manuscript!
> >
> > On the plus side, I really appreciate the additional experiments on datasets with "real" missing values. I also think that the mathematics are significantly clearer (e.g. around Eqn 3). I do not really understand how you compared to Neumiss though, since it is, to the best of my understanding, a regression method, and you are mostly interested in classification.
> >
> > On the minus side, I think that the fact that the algorithm that not necessarily complies with the constraints on $K_\Delta^*$ is concerning, and I don't think the consequences of this are as clear as the authors suggest. Indeed, one can still clip the values again, but that might break positive definiteness. One could repeat the operations many times, in an alternating projection manner, but that would lead to a different algorithm.
> >
> > Two side-notes:
> >
> > - I am still puzzled that MI appears to be the strongest baseline in general (including your new experiments). It would be worth investigating this more, and study for instance if this is indeed because of the influence of the test features, as you suggest. Such a result is interesting in itself, and would essentially imply that previous methods are not really worth it.
> >
> > - As I re-read the paper, I had the following additional question: it is assumed that the data are normalized in $[0,1]$. How is this done in practice, do you do it before or after removing the missing values, and do you use the test set?

---

> ### Author Response · Authors · 2023-11-23
>
> Thank you for your recognition of the modifications we made and for providing valuable feedback once again. We have addressed your questions as follows:
>
> **R1.7 Classification/Regression in NeuMiss**
>
> Firstly, you are absolutely right that NeuMiss (at least in their paper) is designed for regression problems with missing data. To fit the classification task, in NeuMiss, we utilize a certain threshold to convert the soft labels into binary labels 0 and 1. When using such weaker supervised information compared to regression, our algorithm yields better results than NeuMiss. In the future, we are certainly dedicated to extending the ideas presented in this paper to regression tasks, such as utilizing support vector regression models. We will continue exploring this direction in future research.
>
> **R1.8 Further discussion on constraints on $K_\Delta^\ast$**
>
> For optimization problem (5), we have two concerns: 1. The constraint of the matrix being PSD. This is a very hard constraint as it is a prerequisite for training SVM models effectively. 2. The constraint of the matrix falling within a proper range. Roughly speaking, this constraint acts more like regularization, as described in **R1.4(c)**, aiming to limit excessive variations in the optimization variables. Therefore, we adopt a two-step approach of clipping and then projecting to ensure that the first constraint is satisfied while attempting to satisfy the second constraint as much as possible. Indeed, as you mentioned and as we explained in our previous reply, this cannot guarantee that the second constraint is satisfied. Regarding the idea of alternating projection you mentioned, it is indeed a good idea. However, considering the speed of the algorithm and the relatively lower importance of the second constraint, we have opted for the current approach.
>
> **R1.9 Further discussion on results of MI**
>
> Firstly, our algorithm and several other methods do not utilize test set feature information, while MI does. Therefore, in cases where the test set does not have missing values, this additional information becomes a strong prior for the entire data set, resulting in a significant improvement in the performance of MI. Due to time limitations, we conducted experiments with MI excluding test set feature information on the german dataset with $m=80$% and the cylinder dataset. The former yielded a result of $0.686 \pm 0.030$, and the latter resulted in $0.620 \pm 0.024$. Comparing these results with the results of $0.700 \pm 0.020$ and $0.648 \pm 0.014$ in Table 3 and Table 5, respectively, it can be observed that the performance of MI decreases to some extent when lacking the assistance of abundant information that originally exists in the test set. In practice, scenarios with or without missing values in the test set can occur. Therefore, we still highly value and consider the ideas and techniques provided by previous methods to be important.
>
> **R1.10 The way of normalization**
>
> This operation is performed before removing the missing values and does not involve the use of test set data.
>
> It is worth noting that even with a high rate of missing data, we are still able to perform normalization based on the observed features.

---

> > ### Comment · Reviewer_Jet1 · 2023-11-23
> >
> > Thank for you very quick response. I have a few additional comments and questions:
> >
> > About Neumiss:
> >
> > It appears to be a method to perform linear regression with missing values (using a neural net, but the end-goal is to learn a linear model), so I really do not think it is a suitable baseline to compare it against non-linear classification methods. There are many good baselines for supervised learning with missing values, for instance the "HistGradientBoostingClassifier" from sklearn, or even xgboost can handle missing values in an efficient way. So if an additional baseline has to be added, I think one of these would be a better fit than Neumiss. It is true that the baselines implemented by the authors are quite weak (and outperformed by mean imputation), so maybe it would be nice to add such a strong baseline. Although I'd be more interested in understanding why other baselines are so weak.
> >
> > About normalization:
> >
> > I don't understand, you say that it is "performed before removing the missing values", and then that you are "still able to perform normalization based on the observed features".
> >
> > What data set do you feed the normaliser, the complete one (train + test), the complete one (just train), the incomplete one (train + test), or the incomplete one (just train ?). It is important, because this will affect the range of the values, which is critical for your constraints.

---

### Author Response · Authors · 2023-11-17
**Auther Rebuttal**

Dear Program Chairs, Area Chairs, and Reviewers,

First of all, we would like to thank you for your time, constructive critiques, and valuable suggestions, which have greatly helped us improve our work. We appreciate the recognition of the novelty of our proposed two-stage algorithm by reviewers Jet1, h2QN, and VYgf. We would like to thank reviewers h2QN, VYgf, and UxXs for acknowledging our writing in the manuscript. Additionally, we extend our gratitude to reviewers Jet1 and UxXs for their meticulous feedback on unclear parts of our manuscript. We will now summarize the questions raised by all reviewers and provide our responses as follows.

**R0.1 Additional comparison methods and different missing mechanisms.**

We appreciate the valuable suggestions provided by the reviewers, including testing data with inherent missingness (non-MCAR) and comparing against recent popular methods in supervised learning with missing data. This will significantly contribute to evaluating the effectiveness of our approach. As suggested by reviewer Jet1, h2QN and UxXs，we selected NeuMiss [1, 2], a neural network-based completion method, as an additional comparison method. We evaluated the performance of each method on two datasets, horse [3] and cylinder [4], which inherently contain missing values (where the missing values do not follow MCAR). The Horse dataset consists of 368 instances and 25-dimensional features, with approximately 30% of the features missing. The Cylinder dataset comprises 512 instances and 35-dimensional features, with approximately 10% of the features missing. The results are presented in Table 0.1.

Table 0.1: Classification accuracy (mean ± std) on two non-MCAR data sets.
| Data Sets |         MI        |        GEOM       |       KARMA       |       genRBF      |      NeuMiss      |        Ours       |
|:---------:|:-----------------:|:-----------------:|:-----------------:|:-----------------:|:-----------------:|:-----------------:|
|   Horse   | 0.844 $\pm$ 0.027 | 0.848 $\pm$ 0.021 | 0.842 $\pm$ 0.020 | 0.836 $\pm$ 0.028 | 0.843 $\pm$ 0.027 | 0.862 $\pm$ 0.029 |
|  Cylinder | 0.648 $\pm$ 0.014 | 0.620 $\pm$ 0.021 | 0.678 $\pm$ 0.018 | 0.620 $\pm$ 0.021 | 0.625 $\pm$ 0.051 | 0.672 $\pm$ 0.020 |

On these two non-MCAR data sets, our algorithm also demonstrates excellent classification performance. On the Horse dataset, our accuracy is significantly better than other baseline methods according to the paired t-test at the 5% significance level. For the Cylinder data set, predicting for this data set is challenging for all algorithms, and our algorithm performs slightly lower than KARMA but significantly higher than other methods. Furthermore, the suggested method NeuMiss, though a well-designed neural network-based method, exhibits unsatisfactory performance on small data sets. This further highlights the significance of the kernel-based method we proposed.

**R0.2 Discussions on the application on large-scale data.**

We appreciate the interesting question about scalibility. We would like to emphasize that our algorithm focuses on utilizing pairwise information between the data points to guide the imputation process. As you observed in **R0.1**, it is precisely the exploration of this information that enables us to achieve even better results than neural network-based methods in small data sets scenarios. Admittedly, dealing with an $N \times N$ matrix is inevitable in our algorithm, where $N$ denotes the number of data points. This limitation hinders the effective scalability of our algorithm. Furthermore, as you may know, there are methods such as Random Fourier Feature (RFF) and Nyström methods that can accelerate kernel-based algorithms. However, these methods are not suitable for imputation tasks. For instance, RFF fails to compute the inner product between random features $w$ and data $x$ when dealing with missing data. Of course, exploring how to leverage these methods or develop new techniques to extend our algorithm to larger-scale data scenarios will be a worthwhile endeavor for future research.

Once again, we sincerely appreciate your insightful suggestions. We will incorporate the relevant results and discussions into our manuscript. We will continuously revise the paper and upload the revised version later. We look forward to further discussions with you.

Best wishes,

Anonymous author(s) of Paper7804

**Reference**

[1] Le Morvan M, Josse J, Moreau T, et al. NeuMiss networks: differentiable programming for supervised learning with missing values[J]. Advances in Neural Information Processing Systems, 2020, 33: 5980-5990.

[2] Le Morvan M, Josse J, Scornet E, et al. What’s a good imputation to predict with missing values?[J]. Advances in Neural Information Processing Systems, 2021, 34: 11530-11540.

[3] McLeish,Mary and Cecile,Matt. (1989). Horse Colic. UCI Machine Learning Repository.

[4] Evans,Bob. (1995). Cylinder Bands. UCI Machine Learning Repository.

---

### Author Response · Authors · 2023-11-22

Dear Program Chairs, Area Chairs, and Reviewers,

We have resubmitted a revised version of the paper and would appreciate any valuable feedback from you. We will commit to making continuous improvements based on your suggestions and aim to present the best possible content in the final version.

Best wishes,

Anonymous author(s) of Paper7804

---

### Meta-Review · Area_Chair_VuvP · 2023-12-07

**Metareview:**

This submission contributes a supervised method to optimize imputation of missing values in kernel machines. The submission sparked interest from reviewers. The contribution was seen as interesting. However, it was seen as on the light side for ICLR. Indeed, the reviewers pointed out the weakness of the empirical evidence, coming only for 4 small datasets and with unclear statistical significance, as well as the lack of solid theoretical study. In addition, the method is applicable only to kernels and thus in kernel machines, which limits its use.

**Justification For Why Not Higher Score:**

The experiments are very light, the technique applies only to kernel machines, and the theoretical properties are not studied.

One reviewer raised a lot the rating after discussion with authors, but the points he originally raised are still present.

**Justification For Why Not Lower Score:**

N/A

---

### Decision · Program_Chairs · 2024-01-16

Reject